# Sociodemographic and lifestyle predictors of incident hospital admissions with multimorbidity in a general population, 1999–2019: the EPIC-Norfolk cohort

Robert Luben [ORCID],[1] Shabina Hayat,[1] Nicholas Wareham,[2] Paul P Pharoah,[1] Kay-Tee Khaw[2]

[1]Department of Public Health and Primary Care, University of Cambridge School of Clinical Medicine, Cambridge, UK
[2]MRC Epidemiology Unit, University of Cambridge School of Clinical Medicine, Cambridge, UK

**Correspondence to**
Dr Robert Luben;
robert.luben@phpc.cam.ac.uk

## ABSTRACT

**Background** The ageing population and prevalence of long-term disorders with multimorbidity are a major health challenge worldwide. The associations between comorbid conditions and mortality risk are well established; however, few prospective community-based studies have reported on prior risk factors for incident hospital admissions with multimorbidity. We aimed to explore the independent associations for a range of demographic, lifestyle and physiological determinants and the likelihood of subsequent hospital incident multimorbidity.

**Methods** We examined incident hospital admissions with multimorbidity in 25 014 men and women aged 40–79 in a British prospective population-based study recruited in 1993–1997 and followed up until 2019. The determinants of incident multimorbidity, defined as Charlson Comorbidity Index ≥3, were investigated using multivariable logistic regression models for the 10-year period 1999–2009 and repeated with independent measurements in a second 10-year period 2009–2019.

**Results** Between 1999 and 2009, 18 179 participants (73% of the population) had a hospital admission. Baseline 5-year and 10-year incident multimorbidities were observed in 6% and 12% of participants, respectively. Age per 10-year increase (OR 2.19, 95% CI 2.06 to 2.33) and male sex (OR 1.32, 95% CI 1.19 to 1.47) predicted incident multimorbidity over 10 years. In the subset free of the most serious diseases at baseline, current smoking (OR 1.86, 95% CI 1.60 to 2.15), body mass index >30 kg/m² (OR 1.48, 95% CI 1.30 to 1.70) and physical inactivity (OR 1.16, 95% CI 1.04 to 1.29) were positively associated and plasma vitamin C (a biomarker of plant food intake) per SD increase (OR 0.86, 95% CI 0.81 to 0.91) inversely associated with incident 10-year multimorbidity after multivariable adjustment for age, sex, social class, education, alcohol consumption, systolic blood pressure and cholesterol. Results were similar when re-examined for a further time period in 2009–2019.

**Conclusion** Age, male sex and potentially modifiable lifestyle behaviours including smoking, body mass index, physical inactivity and low fruit and vegetable intake were associated with increased risk of future incident hospital admissions with multimorbidity.

## Strengths and limitations of this study

► We examined future hospital admission with multimorbidity using a prospective design and a community-based population.
► The relationship between demographic, lifestyle and physiological factors and subsequent multimorbidity was documented.
► Measurements were made at two time-points: in mainly middle-aged participants (40–79 years) and mainly old-aged participants (48–92 years).
► Participants were followed over 20 years, allowing several time periods to be examined.
► Restricting the definition of multimorbidity to a subset of chronic conditions means some conditions will not be counted.

## INTRODUCTION

The Academy of Medical Sciences 2018 report highlighted multimorbidity as a global priority for research. Patients with multimorbidity experience reduced well-being and quality of life and account for a disproportionately high share of healthcare workload and costs. Management of the rising prevalence of long-term disorders is the main challenge facing healthcare systems worldwide.[1–3]

Multimorbidity is commonly defined as the presence of multiple diseases or conditions with a cut-off of two or more conditions[4]; however, there is no agreed definition or classification system, which makes the existing evidence base difficult to interpret.[1] The term comorbidity predates multimorbidity and was used to predict the effect of additional diseases for those with an index disease of interest.[5–7] The Charlson Comorbidity Index (CCI)[8] was originally created to predict mortality in hospital patients after 1 year and is defined using a set of 17 chronic diseases,

weighted according to the risk of death. The index has been widely used, with several authors suggesting extensions or modifications to the original definition,[9–14] and it remains a common standard with which other systems are often compared.[15]

The associations between comorbid conditions and mortality are well established.[16–20] However, few studies have examined the determinants of incident multimorbidity rather than its consequences,[21–26] since most lack detailed demographic, socioeconomic and physiological measurements in population-based men and women prior to the onset of multimorbid disease with subsequent follow-up. Retrospective hospital-based studies examining multimorbidity lack community-based denominators, while general practice-based studies are often cross-sectional or examine mortality in already multimorbid patients. Few studies examine factors that predict the likelihood of multimorbidity rather than factors that predict risk of individual component conditions. The large majority of studies conducted to date are cross-sectional, with few prospective community-based studies able to examine incident multimorbidity from subsequent hospitalisation.[1 24 25] In this study, we examine the independent associations for a range of demographic, lifestyle and physiological determinants and the likelihood of subsequent hospital incident multimorbidity. We use the CCI over 5-year and 10-year time periods and re-examine these associations independently in a subset 12 years after baseline since healthcare policy and the criteria used for admission may have changed over time. We have previously reported on risk factors for hospitalisation,[27–29] but here we explore in more detail hospital admissions with multimorbidity, a measure of both health service and individual burden.

## METHODS

We used data from the European Prospective Investigation into Cancer in Norfolk cohort (EPIC-Norfolk).[30 31] From this cohort, 25 639 men and women aged 40–79 were recruited from general practices in Norfolk, completed a lifestyle questionnaire and attended a baseline health check from 1993 to 1997. Participants were reapproached approximately 12 years later, aged 48–92, with 9814 completing a second questionnaire and 8049 attending a health check at time-point 2 (TP2). Figure 1 shows a flow diagram of the number of participants at various stages. The cohort was followed until 2019 with annual record linkage to hospital episode data. Since linkage was to national databases and migration of cohort participants was rare, there was almost no loss to follow-up.

The CCI is defined using a set of chronic diseases, each having an associated weight (1, 2, 3 or 6) related to the risk of death. The conditions are myocardial infarction, congestive heart failure, peripheral vascular disease, cerebrovascular disease, dementia, chronic pulmonary disease, rheumatoid disease, peptic ulcer disease, liver disease, diabetes, hemiplegia or paraplegia, renal disease,

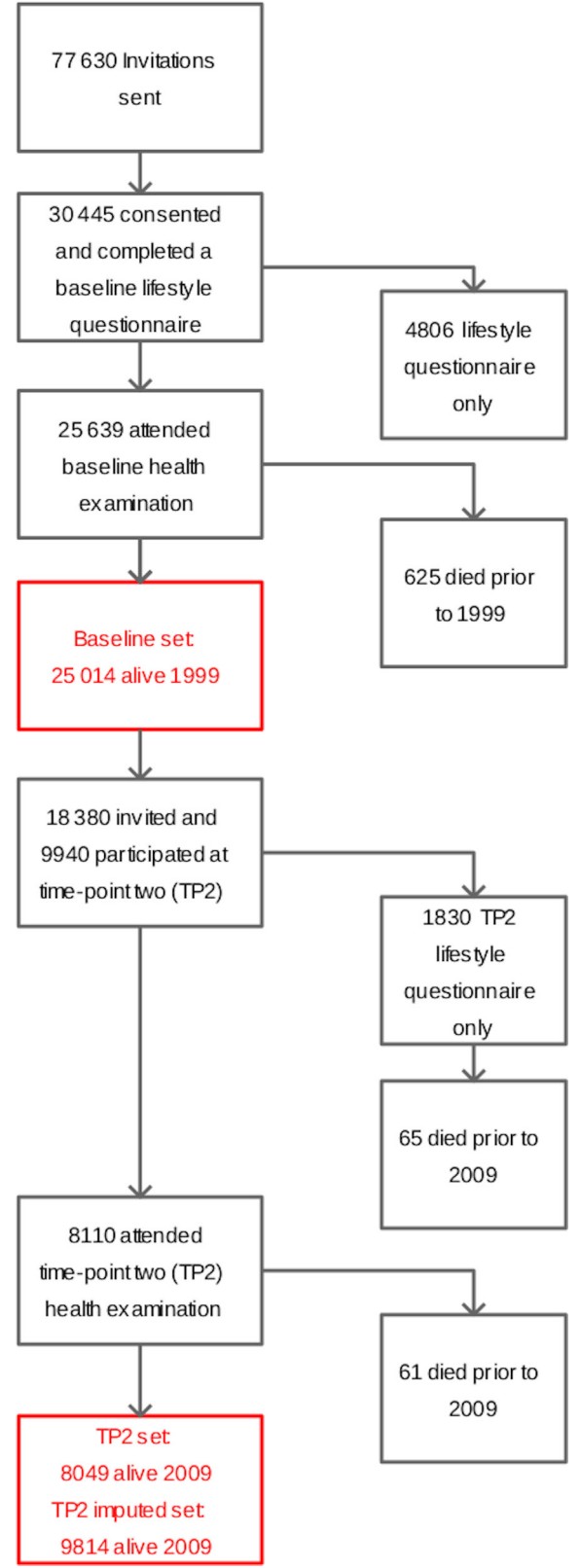

**Figure 1** Flow diagram of cohort recruitment and approaches.

cancer and AIDS/HIV. Two levels of severity are defined for liver disease, diabetes and cancer (details are shown in online supplemental table S1). All comorbidities are assigned a weight of 1, except hemiplegia/paraplegia,

renal disease and malignancies (weight=2); moderate/severe liver disease (weight=3); and metastatic solid tumour and AIDS/HIV (weight=6). For diseases with two levels of severity (liver disease, diabetes and cancer), the less severe version is assigned a weight of 0 if the more severe version is also present in a patient. The CCI diseases were assigned diagnosis codes using the International Classification of Diseases (ICD-10), which was used to link the CCI to Hospital Episode Statistics records and to cohort participants. The weighted individual disease scores were totalled to create an overall score with a maximum value of 29.[8 12] CCI was measured for various outcome periods restricted to all hospital events within the given time period: at baseline, 5-year (1999–2004) and 10-year (1999–2009) CCI; and at TP2, 5-year (2009–2014) and 10-year (2009–2019) CCI. Multiple admissions including the same CCI category were only counted once.

Participants attending the baseline and TP2 health examinations had their height to the nearest 0.1 cm measured using a stadiometer (Chasemores, UK) and their weight to the nearest 100 g measured in light clothing without shoes (Salter, West Bromwich, UK). Body mass index (BMI) was calculated using measured weight in kilograms divided by the square of measured height in square metres. Trained nurses obtained non-fasting blood samples by venepuncture into plain and citrate bottles. Bloods were assayed at the Department of Clinical Biochemistry, University of Cambridge, UK. Serum concentrations of total cholesterol were measured with the RA-1000 Technicon analyser (Bayer Diagnostics, Basingstoke). Plasma was stabilised in a standardised volume of metaphosphoric acid stored at −70°C and vitamin C concentrations measured using a fluorometric assay within 1 week.[32] Systolic blood pressure was measured using an Accutorr sphygmomanometer (Datascope Medical, Huntington, UK). Participants sat for 3 min before two measurements were taken with the arm horizontal and held at mid-sternum level. Systolic blood pressure was defined as the average of the two measurements.

At baseline and again at TP2, participants completed a lifestyle questionnaire. Two yes/no questions were used to derive smoking status: 'Have you ever smoked as much as one cigarette a day for as long as a year?' and, where a positive response was given, 'Do you smoke cigarettes now?' Participants also completed questions about their employment and that of their partner, with details of both current and past employment recorded. Occupational social class was defined according to the Registrar General's classification.[33 34] A list of common UK qualifications was used to establish educational attainment and participants were asked to mark all relevant qualifications. These were then categorised using the highest qualification attained. Participants were asked about their occupational and leisure physical activity. A combined score was created combining leisure and occupational elements and divided into four ordered categories, with those who did not complete the question placed in the inactive category. The score was validated against energy expenditure measured by free-living heart rate monitoring with individual calibration.[35 36] Participants were asked 'Are you a non-drinker/teetotaller now?' and 'At present, about how many alcoholic drinks do you have each week' for various types of alcohol. Current units were calculated from the questionnaire responses, with one unit equal to a half pint of beer, one glass of wine or fortified wine or a single measure of spirits. Prevalent disease was established from the question 'Has the doctor ever told you that you have any of the following?' followed by a list of common conditions including 'Heart attack (myocardial infarction)', 'Stroke', 'Cancer' and 'Diabetes'.

## Statistical methods

Associations were examined both including and excluding chronic disease at baseline and repeated with independent measurements at TP2 in a subset of participants using a second baseline 12 years approximately after the first. The baseline analysis excludes 625 men and women who died before 1999, while at TP2 a further 126 participants who died prior to 2009 were excluded. Dichotomous variables were created for social class (manual and non-manual), educational attainment (high and low) at baseline, and BMI (>30 kg/m² and ≤30 kg/m²) and usual physical activity (active and inactive) at both baseline and TP2. For social class, professional, managerial and technical and non-manual skilled occupations were classed as non-manual, while manual skilled, partly skilled and unskilled were classed as manual. For educational attainment, those with qualifications at secondary level or above were classed as high and those with no qualification as low. Hospital outcomes were categorised into five groups: 'No hospital admissions', CCI=0, CCI=1, CCI=2 and hospital admissions with multimorbidity (incident multimorbidity) defined as CCI ≥3. Multivariable logistic regression was used for all models and compared multimorbid participants (CCI ≥3) with those having CCI ≤2 or no hospital admissions. A sensitivity analysis, using identical models to those in the primary analyses for the period 1999–2009, but excluding 80 participants defined as multimorbid having only one condition with a CCI weighting ≥3, gave virtually identical results (results not shown).

The numbers of individuals with missing values for covariables at baseline were 53 for BMI, 218 for smoking status, 545 for social class and 18 for education level. The physical activity score has no missing values since those with missing data were classified as being inactive. Multiple imputation was used to estimate missing values at TP2 most apparent when participants completed questionnaires but did not attend a health examination (n=1891). Predictive mean matching with 5 multiple imputations and 50 iterations was used with baseline and TP2 variables. All analyses were performed using the R statistical language (V3.5.3, R Foundation for Statistical Computing, Vienna, Austria, with packages knitr, Gmisc, ggplot2, tidyverse, intubate, mice). CCIs were calculated using the R package 'comorbidity'.[37]

**Table 1**  Charlson Comorbidity Index (CCI) hospital admission rates by age group and sex in men and women aged 40–79, 1999–2019

| | Total | No admissions | CCI=0 | CCI=1 | CCI=2 | CCI ≥3 |
|---|---|---|---|---|---|---|
| Baseline 5-year follow-up period, 1999–2004, n (%) | | | | | | |
| Men | 11228 | 5457 (48.6) | 3340 (29.7) | 988 (8.8) | 662 (5.9) | 781 (7.0) |
| Women | 13786 | 7153 (51.9) | 4398 (31.9) | 953 (6.9) | 643 (4.7) | 639 (4.6) |
| ≤55 years | 9567 | 6009 (62.8) | 2720 (28.4) | 411 (4.3) | 236 (2.5) | 191 (2.0) |
| 55–65 years | 7805 | 3940 (50.5) | 2479 (31.8) | 583 (7.5) | 408 (5.2) | 395 (5.1) |
| 65–75 years | 6933 | 2489 (35.9) | 2322 (33.5) | 830 (12.0) | 561 (8.1) | 731 (10.5) |
| >75 years | 709 | 172 (24.3) | 217 (30.6) | 117 (16.5) | 100 (14.1) | 103 (14.5) |
| Baseline 10-year follow-up period, 1999–2009, n (%) | | | | | | |
| Men | 11228 | 2928 (26.1) | 4151 (37.0) | 1434 (12.8) | 1056 (9.4) | 1659 (14.8) |
| Women | 13786 | 3907 (28.3) | 5767 (41.8) | 1601 (11.6) | 1137 (8.2) | 1374 (10.0) |
| ≤55 years | 9567 | 3720 (38.9) | 4201 (43.9) | 746 (7.8) | 476 (5.0) | 424 (4.4) |
| 55–65 years | 7805 | 1973 (25.3) | 3259 (41.8) | 994 (12.7) | 711 (9.1) | 868 (11.1) |
| 65–75 years | 6933 | 1059 (15.3) | 2294 (33.1) | 1168 (16.8) | 875 (12.6) | 1537 (22.2) |
| >75 years | 709 | 83 (11.7) | 164 (23.1) | 127 (17.9) | 131 (18.5) | 204 (28.8) |
| Time-point 2, 5-year follow-up period, 2009–2014, n (%) | | | | | | |
| Men | 4252 | 1428 (33.6) | 1355 (31.9) | 522 (12.3) | 389 (9.1) | 558 (13.1) |
| Women | 5562 | 2234 (40.2) | 1793 (32.2) | 686 (12.3) | 403 (7.2) | 446 (8.0) |
| ≤55 years | 342 | 215 (62.9) | 92 (26.9) | 19 (5.6) | 10 (2.9) | 6 (1.8) |
| 55–65 years | 3090 | 1540 (49.8) | 1006 (32.6) | 277 (9.0) | 143 (4.6) | 124 (4.0) |
| 65–75 years | 3695 | 1303 (35.3) | 1301 (35.2) | 464 (12.6) | 286 (7.7) | 341 (9.2) |
| >75 years | 2687 | 604 (22.5) | 749 (27.9) | 448 (16.7) | 353 (13.1) | 533 (19.8) |
| Time-point 2, 10-year follow-up period, 2009–2019, n (%) | | | | | | |
| Men | 4252 | 695 (16.3) | 1294 (30.4) | 631 (14.8) | 558 (13.1) | 1074 (25.3) |
| Women | 5562 | 1166 (21.0) | 1956 (35.2) | 914 (16.4) | 618 (11.1) | 908 (16.3) |
| ≤55 years | 342 | 154 (45.0) | 122 (35.7) | 37 (10.8) | 14 (4.1) | 15 (4.4) |
| 55–65 years | 3090 | 905 (29.3) | 1241 (40.2) | 407 (13.2) | 267 (8.6) | 270 (8.7) |
| 65–75 years | 3695 | 589 (15.9) | 1309 (35.4) | 611 (16.5) | 473 (12.8) | 713 (19.3) |
| >75 years | 2687 | 213 (7.9) | 578 (21.5) | 490 (18.2) | 422 (15.7) | 984 (36.6) |

## RESULTS

Table 1 shows future 5-year and 10-year CCI hospital admission rates from baseline for 25014 and from TP2 for 9814, according to demographic characteristics in the study population. Between 1999 and 2009, 18179 participants (73% of the population) had a hospital admission. Baseline 5-year and 10-year incident multimorbidities (CCI ≥3) were observed in 6% and 12% of participants, respectively. Figure 2 shows the 10-year multimorbidity rates by age group and sex excluding those with cardiovascular disease, cancer or diabetes at baseline. More men had CCI ≥3 than women and those aged >75 years had the highest proportion of admissions with multimorbid conditions, with 14.5% at 5 years and 28.8% at 10 years. Multimorbidity rates at TP2 were slightly higher than baseline, with 5-year and 10-year incident CCI ≥3 observed in 10% and 20% of participants, respectively, and the highest proportion in those >75 years.

Descriptive characteristics of the cohort according to 10-year CCI are shown in table 2. Participants with higher number of total admissions and longer duration of hospital stay had higher CCI, with mean duration of 58 days and 13 admissions for participants with CCI ≥3 during the 10-year period. Participants with multimorbidity admissions were more likely at baseline examination to be current smokers, less physically active, have higher BMI and have lower plasma vitamin C (a proxy for a diet rich in fruit and vegetables) and report various prevalent conditions.

In table 3, ORs are shown for 5-year and 10-year incident multimorbidity, defined as those with CCI ≥3, compared with CCI ≤2 or no hospital admission, adjusted for age, sex, occupational social class and educational attainment in model 1. Model 2 additionally adjusted for prevalent diseases, cardiovascular disease (CVD), cancer and diabetes; model 3 added lifestyle factors, current

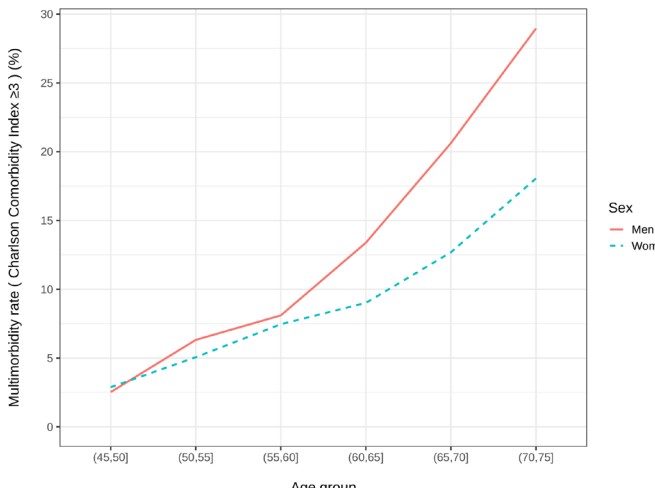

**Figure 2** Rate of hospital admissions with multimorbidity, defined as Charlson Comorbidity Index ≥3, by age group and sex, over the 10-year follow-up period 1999–2009, excluding those with cardiovascular disease, cancer or diabetes at baseline.

smoking, alcohol units per week, usual physical activity as well as BMI >30 kg/m² and plasma vitamin C; and model 4 added systolic blood pressure and cholesterol. Age, sex and prevalent diseases were strongly associated with multimorbidity admissions in all models. The fully adjusted association of 10-year incident multimorbidity with age per 10-year increase had OR of 2.19 (95% CI 2.06 to 2.33), OR of 1.32 (95% CI 1.19 to 1.47) for sex, OR of 2.22 (95% CI 1.87 to 2.62) for prevalent CVD, OR of 2.05 (95% CI 1.73 to 2.42) for cancer, and OR of 3.41 (95% CI 2.74 to 4.24) for diabetes. The risk of multimorbidity in participants with CVD at baseline was equivalent to the risk in those without CVD 10 years older. Similarly, in participants with baseline diabetes and baseline cancer, the risk was equivalent to those without disease aged 17 and 11 years older, respectively.

The models in table 4 are similar to those used in table 3, but rather than adjusting for prevalent disease, participants who reported heart attack, stroke, cancer or diabetes at baseline were excluded. In this subgroup of participants without known common major diseases, in addition to age and sex, current cigarette smoking (OR 1.86, 95% CI 1.60 to 2.15), BMI >30 kg/m² (OR 1.48, 95% CI 1.30 to 1.70) and physical inactivity (OR 1.16, 95% CI 1.04 to 1.29) were positively associated and plasma vitamin C (OR 0.86, 95% CI 0.81 to 0.91) inversely associated with incident 10-year hospital admissions with multimorbidity after multivariable adjustment for age, sex, social class, education, alcohol consumption, systolic blood pressure and cholesterol (model 3). Manual social class and educational attainment were associated with incident multimorbidity in model 1, but were attenuated in models 2 and 3. An inverse association was observed for total cholesterol, while systolic blood pressure appeared to be associated but the direction of association was not consistent with the repeated analyses from TP2. There was no association

for alcohol in these models. The risk of multimorbidity in current cigarette smokers is equivalent to the risk in non-smokers 7 years older, while each 20 µmol/L rise in plasma vitamin C (approximately two servings of fruit and vegetables per day[38]) corresponds to a reduction in risk equivalent to the risk of those 3 years younger.

Online supplemental table S1 shows the ICD-10 codes corresponding to CCI disease groups. Online supplemental table S2 shows the descriptive characteristics of participants at TP2 for 10-year CCI. The mean age in this subset, measured approximately 12 years after baseline, was 69.4. The number of hospital admissions and total length of stay were similar to those at baseline, with multimorbid participants (CCI ≥3) having much longer duration than non-multimorbid participants or those who had no hospital admissions. Multimorbid participants were inactive, had lower plasma vitamin C (reflecting a lower intake of fruit and vegetables), were current or former smokers, and had prevalent disease. In online supplemental table S3, multivariable models of 10-year incident multimorbidity show that prevalent diabetes, CVD and cancer were all strongly associated. online supplemental table S4 shows multivariable associations in a group free from the most serious diseases at TP2. Both age and male sex were associated with subsequent multimorbidity, with educational attainment, current cigarette smoking, plasma vitamin C, BMI >30 kg/m² and physical inactivity all predicting future multimorbidity. Systolic blood pressure was attenuated while other factors including cholesterol were more strongly associated than at baseline.

## DISCUSSION

In this community-based population followed prospectively, we observed incident hospital multimorbidity admission rates over 5-year and 10-year periods, which as expected were strongly related to increasing age. We also observed that those with multimorbid hospital admissions had substantially more days in the hospital over the outcome periods. In multivariable analyses, the risk of such admissions is predicted by age, male sex and several potentially modifiable factors. Participants at baseline who smoked cigarettes, had BMI >30, were physically inactive or had a diet low in fruit and vegetables all had higher likelihood of having subsequent hospital admissions with multimorbidity. Measurements made on a subset of the cohort 12 years after baseline who were followed up subsequently confirmed the baseline findings while also demonstrating an association for low education level in an older cohort with incident multimorbidity.

### Strengths and limitations of the study
Most studies of multimorbidity focus on its consequences and those examining risk factors for multimorbidity are largely cross-sectional. While many prospective studies have examined the relationship between baseline characteristics and specific incident diseases or mortality, establishing multimorbidity as an endpoint is more

**Table 2** Descriptive characteristics at baseline in 25 014 men and women aged 40–79 by 10-year Charlson Comorbidity Index (CCI), 1999–2009

| | Total | No admissions | CCI=0 | CCI=1 | CCI=2 | CCI ≥3 |
|---|---|---|---|---|---|---|
| **Hospital duration 1999–2009, days** | | | | | | |
| Mean±SD | 16.3±46.5 | 0.0±0.0 | 9.1±28.3 | 24.9±71.5 | 30.4±43.0 | 57.8±77.5 |
| **Total hospital admissions 1999–2009** | | | | | | |
| Mean±SD | 3.8±16.2 | 0.0±0.0 | 2.8±3.1 | 4.5±6.0 | 6.4±8.3 | 13.4±43.9 |
| **Age, years** | | | | | | |
| Mean±SD | 59.0±9.3 | 55.4±8.6 | 57.9±8.8 | 62.0±8.8 | 62.9±8.8 | 65.0±8.0 |
| **Body mass index, kg/m²** | | | | | | |
| Mean±SD | 26.4±3.9 | 25.9±3.7 | 26.2±3.8 | 26.8±4.1 | 26.8±4.3 | 27.3±4.2 |
| **Cigarette smoking, n (%)** | | | | | | |
| Current | 2904 | 751 (25.9) | 1008 (34.7) | 410 (14.1) | 291 (10.0) | 444 (15.3) |
| Former | 10 423 | 2558 (24.5) | 4007 (38.4) | 1352 (13.0) | 979 (9.4) | 1527 (14.7) |
| Never | 11 469 | 3476 (30.3) | 4821 (42.0) | 1245 (10.9) | 903 (7.9) | 1024 (8.9) |
| **Social class dichotomised, n (%)** | | | | | | |
| Non-manual | 14 717 | 4400 (29.9) | 5707 (38.8) | 1733 (11.8) | 1256 (8.5) | 1621 (11.0) |
| Manual | 9752 | 2304 (23.6) | 4029 (41.3) | 1214 (12.4) | 886 (9.1) | 1319 (13.5) |
| **Level of education, n (%)** | | | | | | |
| Higher level | 15 866 | 4922 (31.0) | 6333 (39.9) | 1724 (10.9) | 1277 (8.0) | 1610 (10.1) |
| Lower level | 9130 | 1910 (20.9) | 3576 (39.2) | 1310 (14.3) | 916 (10.0) | 1418 (15.5) |
| **Simple physical activity index, n (%)** | | | | | | |
| Inactive | 7559 | 1681 (22.2) | 2666 (35.3) | 1116 (14.8) | 788 (10.4) | 1308 (17.3) |
| Moderately inactive | 7187 | 2084 (29.0) | 2904 (40.4) | 819 (11.4) | 610 (8.5) | 770 (10.7) |
| Moderately active | 5688 | 1708 (30.0) | 2353 (41.4) | 608 (10.7) | 470 (8.3) | 549 (9.7) |
| Active | 4580 | 1362 (29.7) | 1995 (43.6) | 492 (10.7) | 325 (7.1) | 406 (8.9) |
| **Alcohol intake, units per week** | | | | | | |
| Mean±SD | 7.1±9.5 | 7.7±9.6 | 6.9±9.1 | 6.9±9.5 | 6.7±9.8 | 6.8±10.3 |
| **Plasma vitamin C, µmol/L** | | | | | | |
| Mean±SD | 53.5±20.3 | 55.3±19.8 | 55.4±19.9 | 50.5±20.3 | 51.3±20.9 | 47.6±20.6 |
| **Systolic blood pressure, mm Hg** | | | | | | |
| Mean±SD | 135.3±18.3 | 132.4±17.4 | 133.5±17.5 | 138.7±18.7 | 138.6±19.2 | 142.2±19.3 |
| **Total cholesterol, mmol/L** | | | | | | |
| Mean±SD | 6.2±1.2 | 6.1±1.1 | 6.1±1.1 | 6.3±1.2 | 6.2±1.2 | 6.3±1.2 |
| **Prevalent heart attack, n (%)** | | | | | | |
| No reported heart attack | 24 253 | 6745 (27.8) | 9764 (40.3) | 2886 (11.9) | 2097 (8.6) | 2761 (11.4) |
| Self-reported heart attack | 728 | 85 (11.7) | 143 (19.6) | 146 (20.1) | 94 (12.9) | 260 (35.7) |
| **Prevalent stroke, n (%)** | | | | | | |
| No reported stroke | 24 660 | 6786 (27.5) | 9821 (39.8) | 2975 (12.1) | 2151 (8.7) | 2927 (11.9) |
| Self-reported stroke | 329 | 45 (13.7) | 87 (26.4) | 57 (17.3) | 41 (12.5) | 99 (30.1) |
| **Prevalent cancer, n (%)** | | | | | | |
| No reported cancer | 23 688 | 6595 (27.8) | 9449 (39.9) | 2878 (12.1) | 2031 (8.6) | 2735 (11.5) |
| Self-reported cancer | 1301 | 237 (18.2) | 459 (35.3) | 155 (11.9) | 162 (12.5) | 288 (22.1) |
| **Prevalent diabetes, n (%)** | | | | | | |
| No reported diabetes | 24 442 | 6760 (27.7) | 9844 (40.3) | 2941 (12.0) | 2111 (8.6) | 2786 (11.4) |
| Self-reported diabetes | 541 | 71 (13.1) | 61 (11.3) | 90 (16.6) | 81 (15.0) | 238 (44.0) |

**Table 3** Multivariable logistic regression of risk factors for 5-year and 10-year hospital admissions with multimorbidity in 25 014 men and women

| | 5-year multimorbidity*, 1999–2004 OR (95% CI) | P value | 10-year multimorbidity*, 1999–2009 OR (95% CI) | P value |
|---|---|---|---|---|
| Model 1 | | | | |
| Male sex | 1.49 (1.34 to 1.67) | <0.001 | 1.56 (1.44 to 1.69) | <0.001 |
| Age per 10 years | 2.27 (2.13 to 2.44) | <0.001 | 2.34 (2.23 to 2.46) | <0.001 |
| Manual social class | 1.20 (1.07 to 1.35) | 0.002 | 1.22 (1.12 to 1.33) | <0.001 |
| Lower education level | 1.15 (1.02 to 1.30) | 0.023 | 1.19 (1.09 to 1.30) | <0.001 |
| Model 2 | | | | |
| Male sex | 1.39 (1.24 to 1.56) | <0.001 | 1.47 (1.35 to 1.60) | <0.001 |
| Age per 10 years | 2.11 (1.97 to 2.26) | <0.001 | 2.21 (2.10 to 2.32) | <0.001 |
| Manual social class | 1.22 (1.08 to 1.37) | 0.001 | 1.23 (1.13 to 1.34) | <0.001 |
| Lower education level | 1.13 (1.00 to 1.28) | 0.053 | 1.17 (1.07 to 1.28) | <0.001 |
| Prevalent CVD | 2.23 (1.85 to 2.68) | <0.001 | 2.25 (1.93 to 2.60) | <0.001 |
| Prevalent cancer | 2.11 (1.75 to 2.54) | <0.001 | 1.92 (1.65 to 2.22) | <0.001 |
| Prevalent diabetes | 4.41 (3.55 to 5.45) | <0.001 | 4.32 (3.57 to 5.21) | <0.001 |
| Model 3 | | | | |
| Male sex | 1.24 (1.07 to 1.42) | 0.003 | 1.33 (1.20 to 1.47) | <0.001 |
| Age per 10 years | 2.16 (1.99 to 2.34) | <0.001 | 2.29 (2.16 to 2.43) | <0.001 |
| Manual social class | 1.09 (0.95 to 1.25) | 0.214 | 1.17 (1.06 to 1.29) | 0.002 |
| Lower education level | 1.06 (0.92 to 1.21) | 0.447 | 1.08 (0.98 to 1.20) | 0.112 |
| Current smoker | 1.71 (1.42 to 2.05) | <0.001 | 1.73 (1.51 to 1.98) | <0.001 |
| BMI >30 kg/m² | 1.32 (1.12 to 1.56) | <0.001 | 1.45 (1.28 to 1.63) | <0.001 |
| Alcohol intake, units per week | 1.00 (0.99 to 1.01) | 0.872 | 1.00 (1.00 to 1.01) | 0.666 |
| Physically inactive | 1.26 (1.10 to 1.44) | <0.001 | 1.15 (1.04 to 1.26) | 0.006 |
| Plasma vitamin C per SD | 0.81 (0.75 to 0.86) | <0.001 | 0.84 (0.80 to 0.88) | <0.001 |
| Prevalent CVD | 2.02 (1.63 to 2.49) | <0.001 | 2.17 (1.84 to 2.57) | <0.001 |
| Prevalent cancer | 2.22 (1.79 to 2.72) | <0.001 | 2.06 (1.74 to 2.43) | <0.001 |
| Prevalent diabetes | 3.53 (2.73 to 4.52) | <0.001 | 3.54 (2.85 to 4.39) | <0.001 |
| Model 4 | | | | |
| Male sex | 1.23 (1.07 to 1.43) | 0.005 | 1.32 (1.19 to 1.47) | <0.001 |
| Age per 10 years | 2.08 (1.91 to 2.27) | <0.001 | 2.19 (2.06 to 2.33) | <0.001 |
| Manual social class | 1.09 (0.95 to 1.25) | 0.235 | 1.16 (1.05 to 1.28) | 0.004 |
| Lower education level | 1.06 (0.92 to 1.22) | 0.420 | 1.09 (0.99 to 1.21) | 0.091 |
| Current smoker | 1.72 (1.43 to 2.07) | <0.001 | 1.74 (1.52 to 2.00) | <0.001 |
| BMI >30 kg/m² | 1.31 (1.11 to 1.54) | 0.001 | 1.40 (1.24 to 1.58) | <0.001 |
| Alcohol intake, units per week | 1.00 (0.99 to 1.01) | 0.878 | 1.00 (1.00 to 1.01) | 0.800 |
| Physically inactive | 1.25 (1.09 to 1.43) | 0.001 | 1.14 (1.04 to 1.26) | 0.008 |
| Plasma vitamin C per SD | 0.81 (0.76 to 0.87) | <0.001 | 0.85 (0.81 to 0.89) | <0.001 |
| Systolic blood pressure per SD | 1.10 (1.03 to 1.17) | 0.005 | 1.12 (1.07 to 1.18) | <0.001 |
| Total cholesterol per SD | 0.99 (0.92 to 1.05) | 0.690 | 0.99 (0.94 to 1.04) | 0.614 |
| Prevalent CVD | 2.06 (1.66 to 2.54) | <0.001 | 2.22 (1.87 to 2.62) | <0.001 |
| Prevalent cancer | 2.23 (1.80 to 2.75) | <0.001 | 2.05 (1.73 to 2.42) | <0.001 |
| Prevalent diabetes | 3.42 (2.64 to 4.39) | <0.001 | 3.41 (2.74 to 4.24) | <0.001 |

*Charlson Comorbidity Index ≥3 vs Charlson Comorbidity Index ≤2 or no hospital admission.
BMI, body mass index; CVD, cardiovascular disease.

**Table 4** Multivariable logistic regression of risk factors excluding participants with prevalent CVD, cancer or diabetes for 5-year and 10-year hospital admissions with multimorbidity in 22 278 men and women

| | 5-year multimorbidity*, 1999–2004 OR (95% CI) | P value | 10-year multimorbidity*, 1999–2009 OR (95% CI) | P value |
|---|---|---|---|---|
| Model 1 | | | | |
| Male sex | 1.47 (1.29 to 1.68) | <0.001 | 1.52 (1.38 to 1.67) | <0.001 |
| Age per 10 years | 2.19 (2.02 to 2.37) | <0.001 | 2.31 (2.19 to 2.45) | <0.001 |
| Manual social class | 1.23 (1.07 to 1.42) | 0.003 | 1.22 (1.11 to 1.34) | <0.001 |
| Lower education level | 1.20 (1.04 to 1.39) | 0.011 | 1.16 (1.05 to 1.28) | 0.003 |
| Model 2 | | | | |
| Male sex | 1.32 (1.13 to 1.55) | <0.001 | 1.39 (1.24 to 1.55) | <0.001 |
| Age per 10 years | 2.24 (2.05 to 2.46) | <0.001 | 2.40 (2.25 to 2.56) | <0.001 |
| Manual social class | 1.13 (0.96 to 1.32) | 0.131 | 1.17 (1.05 to 1.30) | 0.006 |
| Lower education level | 1.07 (0.91 to 1.25) | 0.416 | 1.05 (0.93 to 1.17) | 0.428 |
| Current smoker | 1.85 (1.50 to 2.26) | <0.001 | 1.84 (1.58 to 2.13) | <0.001 |
| BMI >30 kg/m² | 1.31 (1.07 to 1.58) | 0.006 | 1.53 (1.34 to 1.75) | <0.001 |
| Alcohol intake, units per week | 1.00 (0.99 to 1.01) | 0.789 | 1.00 (1.00 to 1.01) | 0.805 |
| Physically inactive | 1.25 (1.07 to 1.46) | 0.004 | 1.17 (1.05 to 1.31) | 0.005 |
| Plasma vitamin C per SD | 0.82 (0.76 to 0.89) | <0.001 | 0.85 (0.80 to 0.90) | <0.001 |
| Model 3 | | | | |
| Male sex | 1.32 (1.12 to 1.56) | 0.001 | 1.37 (1.22 to 1.54) | <0.001 |
| Age per 10 years | 2.15 (1.95 to 2.37) | <0.001 | 2.30 (2.15 to 2.46) | <0.001 |
| Manual social class | 1.11 (0.95 to 1.31) | 0.178 | 1.15 (1.03 to 1.29) | 0.012 |
| Lower education level | 1.07 (0.91 to 1.26) | 0.383 | 1.05 (0.94 to 1.18) | 0.393 |
| Current smoker | 1.88 (1.52 to 2.30) | <0.001 | 1.86 (1.60 to 2.15) | <0.001 |
| BMI >30 kg/m² | 1.30 (1.07 to 1.58) | 0.007 | 1.48 (1.30 to 1.70) | <0.001 |
| Alcohol intake, units per week | 1.00 (0.99 to 1.01) | 0.828 | 1.00 (0.99 to 1.01) | 0.941 |
| Physically inactive | 1.24 (1.06 to 1.45) | 0.007 | 1.16 (1.04 to 1.29) | 0.009 |
| Plasma vitamin C per SD | 0.83 (0.77 to 0.90) | <0.001 | 0.86 (0.81 to 0.91) | <0.001 |
| Systolic blood pressure per SD | 1.12 (1.03 to 1.21) | 0.005 | 1.13 (1.07 to 1.19) | <0.001 |
| Total cholesterol per SD | 0.98 (0.91 to 1.06) | 0.607 | 0.97 (0.92 to 1.03) | 0.328 |

*Charlson Comorbidity Index ≥3 versus Charlson Comorbidity Index ≤2 or no hospital admission.
BMI, body mass index; CVD, cardiovascular disease.

challenging. By using the CCI to define multimorbidity, we were able to show that the chronic diseases defined by the index had considerably higher average length of stay than other conditions requiring hospitalisation and that length of stay increased with higher CCI score. The current population-based study in a defined community was able to assess incident hospital admissions with multimorbidity to enable estimates of 5-year and 10-year rates by age and sex. We were also able to document the relationship between demographic, lifestyle and physiological factors and subsequent hospitalisations for multimorbidity. The EPIC-Norfolk cohort has been followed for 20 years, enabling us to examine the determinants of multimorbidity at two time-points: in mainly middle-aged participants (40–79 years) and mainly old-aged participants (48–92 years) in a subcohort 12 years later

after major organisational changes had been made to the National Health Service (NHS). We were also able to examine associations with and without excluding participants with known prevalent conditions at baseline.

While not attempting to examine clusters or pathways of chronic disease, we have identified risk factors that predict any hospital admissions with multimorbidity. It is possible that some factors we observed will be more strongly associated with certain combinations of diseases and others less so. However, the burden of resources experienced by hospitals can best be mitigated by early public health advice, prior to the onset of disease if possible, which can only be general in nature. Our findings are in line with current public health advice such as smoking cessation, a diet containing fruit and vegetables and regular exercise and, given the huge additional burden placed on the

NHS by multimorbidity, should further emphasise the need for public health advice and intervention.

Multimorbidity can be defined in a number of ways, such as disease counts or using various indexes.[39] By restricting the definition to a relatively small subset of chronic conditions such as in the CCI, inevitably some conditions will not be counted. It is notable that the CCI does not include depression or mental health, asthma or respiratory diseases, epilepsy, hypothyroidism, musculoskeletal problems or atrial fibrillation, all common in a primary care setting.[40] In addition to the CCI and other commonly used systems,[41] authors have used many other definitions with variable numbers of underlying conditions and hence the prevalence of multimorbidity varies widely. However, CCI is a widely used measure of multimorbidity.

Since the CCI is weighted to predict mortality, it may be better able to assess health service burden than a simple disease count, since procedures required for higher weighted conditions will generally be more costly. However, it may be less effective as an indicator of multiple long-term conditions. Some chronic conditions such as musculoskeletal and mental health diseases not included in the CCI are nevertheless likely to require long-stay inpatient care. However, increasing CCI had longer hospital length of stay in the present study and this has also been reported in several other studies.[42 43] Medical conditions such as obesity have well-established links to many diseases but, as non-diseases, are not included in the CCI. The use of CCI ≥3 to define multimorbidity classifies a small number of participants with one serious disease with a high CCI weight as multimorbid. However, a sensitivity excluding these people gave virtually identical results. Studies examining the longitudinal predictors of future multimorbidity generally rely on self-reported disease, but our study used the CCI from linked hospital medical coding.

When examining the relationship between lifestyle factors and health outcomes, confounding will always be a limitation. Individuals who smoke, are less physically active and eat a poor diet for example are likely to differ from those with a contrasting lifestyle with respect to other factors relating to the likelihood of future multimorbidity, including their age, sex, lifestyle factors examined in this study and others unknown. However, the associations we report were consistent after multivariable adjustment for other factors. Differential mortality is another possible limitation and would occur for any of the factors examined if participants with an apparently unhealthy characteristic were more likely to have died earlier than those with the contrary healthy characteristic and hence were less likely to use hospital services for the full follow-up period. However, the results for the 5-year follow-up period where very few deaths occurred were consistent with the longer 10-year follow-up period. While it is possible that some participants were multimorbid at baseline, we examined those with and without baseline self-reported major chronic disease.

## Comparison with other studies

Estimates of the prevalence of multimorbidity vary widely, partly due to the variety of definitions, number of diseases, weighting and so on used in studies, but range from 55% to 98% in the elderly.[6] Most studies report multimorbidity associated with age and present in more than half of those aged 65 and older.[3 44] Age was strongly associated with future hospitalisation and incident multimorbidity in our study and has been reported to increase hospitalised multimorbidity in elderly patients.[45] Many studies have found that women have a higher rate of multimorbidity than men,[6 44 46–48] but we observed the converse, with male sex strongly predicting future multimorbidity. The use of CCI in the context of prospective hospital admissions rather than cross-sectional multimorbidity in a primary care setting may explain the higher proportion of multimorbid men. Physical-mental comorbidity is reported higher among women in primary care,[49] and mental health, which is not included in the CCI, may be more likely to be treated in a primary care than in an acute hospital setting.

Despite the considerable literature relating to multimorbidity, very few studies have examined the modifiable determinants of incident multimorbidity. Incident cancer and cardiometabolic multimorbidity were examined in a recent multicentre study which included data from the present study[21]; prediagnostic healthy lifestyle behaviours were reported to be inversely associated with the risk. BMI was also reported to be associated with incident cardiometabolic multimorbidity in a pooled analysis of 16 cohort studies.[22] A Finnish study examined incident multimorbidity in both disease-free and those with baseline diabetes and CVD.[24] They reported some similar findings to the present study such as associations with cigarette smoking, physical inactivity and BMI, but associations for low education level and systolic blood pressure were only found in men. Multimorbidity was defined using five common diseases, and time-to-event 10-year follow-up was used rather than a follow-up period approach in this study. Participants in the Finnish cohort were younger than those in EPIC-Norfolk, with the oldest participant 74 years at the end of follow-up against 90 years in the EPIC-Norfolk baseline and 100 years at TP2. Studies using data from an English longitudinal cohort and using self-reported disease counts to define multimorbidity reported associations in physical activity, obesity and low level of wealth and an increased risk of multimorbidity when combined with other lifestyle factors such smoking, obesity and inadequate fruit and vegetable consumption.[25 26] However, they found no association with educational attainment or excess alcohol consumption. Education, which was associated in older participants at TP2 in our study, has been linked to multimorbidity in cross-sectional studies[50] and prospectively.[24] Socioeconomic status was reported to predict the development of multimorbidity throughout the life course in a Scottish longitudinal study.[51] Both educational attainment and occupational social class were attenuated in

our study possibly due to the models including plasma vitamin C, also a marker of socioeconomic status. While smoking was a strong predictor, we did not find an association with alcohol drinking. However, other studies in the literature are inconsistent, with some finding no association with cigarette smoking and alcohol consumption in cross-sectional analyses.[1]

## Generalisability

While hospital admissions with multimorbidity provide an objective indicator of both health service and individual burden of the condition, studies of hospital admissions in many countries are limited by factors relating to differential accessibility to healthcare such as health insurance, income and healthcare policy. Although not entirely free of differential accessibility, the NHS in the UK, with service free at the point of delivery for all residents, provides an opportunity to examine hospitalised multimorbidity with fewer of these constraints. Healthcare policy and criteria for admission change over time, not least in the UK over the 20-year period of this study, so we examined admissions and risk factors for multimorbidity over two independent time periods using new repeated measures and found consistent results.

## CONCLUSIONS AND POLICY IMPLICATIONS

We observed in a long-term population-based study that age, male sex and potentially modifiable factors including smoking, BMI, physical inactivity and a diet low in fruit and vegetables predict future incident hospitalised multimorbidity. Multimorbidity is increasingly common among elderly hospital inpatients due in part to improved efficacy of treatments and drugs. While considerable effort is being focused on the progression, disease clustering and treatment of patients with multimorbidity, there has been less attention on the long-term predictors of future incident multimorbidity. This study suggests that modest difference in lifestyle may have the potential to mitigate the future burden of multimorbidity in the population.

**Acknowledgements** The authors would like to thank the participants, general practitioners and staff of EPIC-Norfolk.

**Contributors** K-TK, NW, SH and RL were involved in the conception and design of the study. RL drafted the manuscript, with support from K-TK and PPP. SH contributed to data interpretation. RL was responsible for external data linkage. SH and RL contributed to data collection and acquisition. All authors read and critically revised the manuscript and approved the final manuscript. RL is the guarantor. The corresponding author attests that all listed authors meet the authorship criteria and that no others meeting the criteria have been omitted.

**Funding** The design and conduct of the EPIC-Norfolk study and the collection and management of the data were supported by programme grants from the Medical Research Council UK (G9502233, G0401527) and Cancer Research UK (C864/A8257, C864/A2883).

**Disclaimer** The sponsors had no role in any of the following: study design, data collection, data analysis, interpretation of data, writing of the article and decision to submit it for publication. All authors are independent of funders and sponsors and had access to all the data.

**Competing interests** RL, SH, K-TK and NW report grants from MRC and CRUK during the conduct of the study.

**Patient and public involvement** Patients and/or the public were not involved in the design, or conduct, or reporting, or dissemination plans of this research.

**Patient consent for publication** Not required.

**Ethics approval** The work was approved by the East Norfolk and Waveney NHS research governance committee (2005EC07L) and the Norfolk research ethics committee (05/Q0101/191). All participants gave informed signed consent for study participation including access to medical records.

**Provenance and peer review** Not commissioned; externally peer reviewed.

**Data availability statement** Data are available upon reasonable request. The authors will make the data set available under a Data Transfer Agreement to any bona fide researcher who wishes to obtain the data set in order to undertake a replication analysis. Although the data set is anonymised, the breadth of the data included and the multiplicity of variables that are included in this analysis file as primary variables or confounding factors mean that provision of the data set to other researchers without a Data Transfer Agreement would constitute a risk. Requests for data sharing/access should be submitted to the EPIC Management Committee (epic-norfolk@mrc-epid.cam.ac.uk).

**ORCID iD**
Robert Luben http://orcid.org/0000-0002-5088-6343

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
