## [Reviewer comments · BMJ Open]

This paper was submitted to a another journal from BMJ but declined for publication following peer review. The authors addressed the reviewers' comments and submitted the revised paper to BMJ Open. The paper was subsequently accepted for publication at BMJ Open.

ARTICLE DETAILS

TITLE (PROVISIONAL)	Sociodemographic and lifestyle predictors of incident hospital admissions with multimorbidity in a general population 1999–2019: the EPIC-Norfolk cohort
AUTHORS	Luben, Robert; Hayat, Shabina; Wareham, Nicholas; Pharoah, Paul; Khaw, Kay-Tee

VERSION 1 - REVIEW

REVIEWER	Freisling, Heinz IARC
REVIEW RETURNED	19-May-2020

GENERAL COMMENTS	In this study, the authors used annual record linkage to hospital episode data among participants of a prospective cohort to define incident multimorbidity. Associations with a range of sociodemographic and lifestyle factors were investigated. This study addresses an increasingly important topic and the manuscript is generally well written. I do have a number of suggestions to improve clarity and interpretability of the work. 1. Please clarify that all participants were free of multimorbidity at baseline, except for the three self-reported major conditions cardiovascular diseases, diabetes, and cancer.2. Indicate the sampling frame of recruitment.3. Is the Charlson Comorbidity Index (CCI) the sum of repeated admissions (i.e. hospital events) that occurred during the 5-year or 10-year period, where any of the CCI conditions were the reason for admission? How complete is the record linkage?4. The CCI has a range from 0 to 29, while the authors only use a fraction of this information, basically 3 categories in descriptive analysis and 0 vs. ≥ 2 to define multimorbidity. My question is why not use a simple disease count with a more complete list of recorded diseases including depression, respiratory diseases, and other. I also question the statement that the CCI is a de facto standard to define multimorbidity. This is possibly the case in clinic, but in etiological research the CCI is rarely used.5. A graph of the study design indicating the time period, hospital admissions, ascertainment of index conditions etc. would be helpful.6. In which way is the weighing of the individual disease scores important for the current analysis (p. 4, lines 21-22)?7. I would be informative to provide descriptive results of the CCI such as inter-decile range and most common combinations of morbidities. This can be presented as a supplementary information.8. How were competing risks due to death or non-CCI diseases accounted for?9. Is the re-assessment of associations at time point 2 truly
--

	independent as the authors claim (p. 3, line 46)? Please clearly state the rationale for this re-assessment. 10. The statistical analysis needs more details: how was the control group defined for 5-year and 10-year multimorbidity. I assume multimorbidity in the multiple logistic regression was defined as a CCI equal or larger than 2 vs. a CCI of 1 or 0 or vs. a CCI of 0? Was the control group for both outcomes the same or were they different? 11. Some relevant references could be added including Kivimäki et al (https://doi.org/10.1016/S2468-2667(17)30074-9), Freisling et al. (PMID: 31918762). 12. Definition of multimorbidity and how odds ratios were estimated are missing in the abstract.
--	--

REVIEWER	Reviewer 2
REVIEW RETURNED	25-May-2020

GENERAL COMMENTS	Manuscript Number: JECH-2020-214389 Sociodemographic and lifestyle predictors of incident hospital admissions with multimorbidity in a general population 1999-2019: the EPIC-Norfolk cohort This is an interesting paper, but I found it difficult to follow at times. The paper addresses a topic of importance to the readership of the Journal of Epidemiology and Community Health. The authors use longitudinal population-based data linked to hospital records to examine the determinants of incident multimorbidity in two 10-year periods. I do think this is a challenging paper to write as the design and analyses are fairly complex. Comments and suggestions for the authors' consideration are below. 1) It was not clear how chronic conditions at baseline were measured from the methods section. It was not until I got to table 1 (i.e. that they were based on self-report) that I realized this. 2) The CCI-based definition of "multimorbidity" may be problematic. Based on the weighting, it appears to be possible that a person could be categorized as having "multimorbidity" with only one condition. If this is the case, this is not the traditional definition of multimorbidity and perhaps another term should be used. If this is not the case, there was not sufficient detail in the methods to understand how multimorbidity was defined. 3) I am assuming that algorithms were applied to the administrative data to identify people who had each of the relevant CCs. Additional information needs to be provided about these algorithms and their validity. 4) A rationale for the specific covariates chosen as well as a short description of how they were operationalized. For example, it was not clear which variables were directly measure (height, weight) vs. self-reported from the paper. As well, some variables, e.g. smoking and physical activity, were not mentioned in the methods at all. 5) The idea of "incident" multimorbidity implies that they did not have multimorbidity at baseline. Although the authors conduct the analysis in those with and without a subset of the self-reported conditions at
--

	baseline, they do not examine participant who did not have multimorbidity using their definition at baseline. 6) In calculating incident multimorbidity it wasn't clear how death or loss to follow-up during the interval was treated. It seems from the change in sample size from baseline to TP2 that there was fairly significant attrition. Would it be possible to utilize the yearly hospital data using survival analysis and include the competing risk of death? 7) The rationale for the 5 models should be described in the methods section as well as the assessment of model fit. 8) In the discussion the authors suggest that the CCI is the most widely used index of multimorbidity and can be considered a de facto standard. Perhaps this should be qualified by the authors as their definition of multimorbidity is not the standard definition (based on the number of chronic conditions) used by most researchers 9) The last paragraph of the discussion prior to the conclusion was very helpful to provide context for examining the two periods. It may be helpful to move some of this motivation to the introduction.
--	--

VERSION 1 – AUTHOR RESPONSE

> Reviewer: 1

>

> Comments to the Author

> In this study, the authors used annual record linkage to hospital episode data among participants of a prospective cohort to define incident

> multimorbidity. Associations with a range of sociodemographic and lifestyle factors were investigated. This study addresses an increasingly important

> topic and the manuscript is generally well written. I do have a number of suggestions to improve clarity and interpretability of the work.

> 1. Please clarify that all participants were free of multimorbidity at

> baseline, except for the three self-reported major conditions cardiovascular diseases, diabetes, and cancer.

Multimorbidity was not available at study baseline. We now mention that it is possible some participants were multimorbid at baseline and that we examine participants with and without baseline major chronic disease (p.14, Strengths and limitations of study)

> 2. Indicate the sampling frame of recruitment.

A flow diagram (Figure 1) shows details of recruitment while we mention in the Methods that participants were recruited from general practices in Norfolk.

> 3. Is the Charlson Comorbidity Index (CCI) the sum of repeated admissions
> (i.e. hospital events) that occurred during the 5-year or 10-year period,
> where any of the CCI conditions were the reason for admission? How complete
> is the record linkage?

Multiple admissions occurred in a follow-up period with the same CCI categories were only counted once. This is now mentioned in method paragraph 2. Record linkage to national database was performed annual from 1999 to 2019 and since participant migration is rare there is almost no loss to follow-up (methods, paragraph 1)

> 4. The CCI has a range from 0 to 29, while the authors only use a fraction of
> this information, basically 3 categories in descriptive analysis and 0 vs.
> ≥ 2 to define multimorbidity. My question is why not use a simple disease
> count with a more complete list of recorded diseases including depression,
> respiratory diseases, and other. I also question the statement that the CCI
> is a de facto standard to define multimorbidity. This is possibly the case in
> clinic, but in etiological research the CCI is rarely used.

This issue is now explained in more detail in the discussion. Multimorbidity is generally defined using an validated system and CCI, designed to predict 1-year mortality, has been reported to predict hospitalisation.

> 5. A graph of the study design indicating the time period, hospital
> admissions, ascertainment of index conditions etc. would be helpful.

A flow diagram (Figure 1) shows details of recruitment

- > 6. In which way is the weighing of the individual disease scores important
- > for the current analysis (p. 4, lines 21-22)?

This issue is now explained more clearly in the discussion. Conditions with high CCI weighting generally correspond to more expensive procedures and longer length of stay.

- > 7. It would be informative to provide descriptive results of the CCI such as
- > inter-decile range and most common combinations of morbidities. This can be
- > presented as a supplementary information.

We have chosen not include this here but we cite our previous work which common diseases resulting in hospital admission have been described.

- > 8. How were competing risks due to death or non-CCI diseases accounted for?

Our study examines the subsequent risk of hospitalisation within a community population broadly comparable to the UK population. We use number of hospital admissions and length of stay over a fixed period which differs from survival analysis which uses age at the earliest admission and cannot account for multiple events over a long period.

- > 9. Is the re-assessment of associations at time point 2 truly independent as
- > the authors claim (p. 3, line 46)? Please clearly state the rationale for
- > this re-assessment.

Details of the health examination and questionnaires completed at time-point 2 are described in detail in the methods section. The rationale for this re-assessment is now explained in the introduction as well as the conclusion.

- > 10. The statistical analysis needs more details: how was the control group
- > defined for 5-year and 10-year multimorbidity. I assume multimorbidity in the
- > multiple logistic regression was defined as a CCI equal or larger than 2 vs.
- > a CCI of 1 or 0 or vs. a CCI of 0? Was the control group for both outcomes
- > the same or were they different?

We now explain more clearly in both the methods section and results how the comparison groups are defined. Additionally, the cutpoint has been altered to $CCI \geq 3$ (compared to $CCI \leq 2$) to address the issue that some participants may be classified as multimorbid while only have a single high-weighted CCI condition. A sensitivity analysis excluding 80 such people is described in the text.

- > 11. Some relevant references could be added including Kivimäki et al
- > ([https://doi.org/10.1016/S2468-2667\(17\)30074-9](https://doi.org/10.1016/S2468-2667(17)30074-9)), Freisling et al. (PMID:
- > 31918762).

These references are now included.

- > 12. Definition of multimorbidity and how odds ratios were estimated are
- > missing in the abstract.
- >

Multimorbidity is now described in the abstract. The abstract mentions the use of

logistic regression.

> Reviewer: 2

>

> Comments to the Author

> Manuscript Number: JECH-2020-214389

>

> Sociodemographic and lifestyle predictors of incident hospital admissions
> with multimorbidity in a general population 1999-2019: the EPIC-Norfolk

> cohort

>

> This is an interesting paper, but I found it difficult to follow at times.
> The paper addresses a topic of importance to the readership of the Journal of

> Epidemiology and Community Health. The authors use longitudinal
> population-based data linked to hospital records to examine the determinants

> of incident multimorbidity in two 10-year periods. I do think this is a
> challenging paper to write as the design and analyses are fairly complex.

>

> Comments and suggestions for the authors' consideration are below.

>

> 1) It was not clear how chronic conditions at baseline were measured from the

> methods section. It was not until I got to table 1 (i.e. that they were
> based on self-report) that I realized this.

We have now included a much more comprehensive methods section describing baseline covariates and self reported disease.

> 2) The CCI-based definition of "multimorbidity" may be problematic. Based on
> the weighting, it appears to be possible that a person could be categorized

> as having "multimorbidity" with only one condition. If this is the case, this
> is not the traditional definition of multimorbidity and perhaps another term

> should be used. If this is not the case, there was not sufficient detail in
> the methods to understand how multimorbidity was defined.

We recognise this as problematic and have now altered the cutpoint to $CCI \geq 3$ (compared to $CCI \leq 2$) to address this issue. A small number of participants ($n=80$ over the period 1999-2009) are still be classified as multimorbid while only have a single high-weighted CCI condition but a sensitivity analysis excluding these people gave virtually identical results in multivariable models. This is now described in the text.

> 3) I am assuming that algorithms were applied to the administrative data to
> identify people who had each of the relevant CCs. Additional information

> needs to be provided about these algorithms and their validity.

A much more comprehensive methods section is now included which includes references to papers defining the CCI.

- > 4) A rationale for the specific covariates chosen as well as a short
- > description of how they were operationalized. For example, it was not clear
- > which variables were directly measure (height, weight) vs. self-reported from
- > the paper. As well, some variables, e.g. smoking and physical activity, were
- > not mentioned in the methods at all.

A much more comprehensive methods section is now included.

- > 5) The idea of “incident” multimorbidity implies that they did not have
- > multimorbidity at baseline. Although the authors conduct the analysis in

- > those with and without a subset of the self-reported conditions at baseline,
- > they do not examine participant who did not have multimorbidity using their
- > definition at baseline.

Multimorbidity was not available at study baseline. We now mention that it is possible some participants were multimorbid at baseline and that we examine participants with and without baseline major chronic disease (p.14, Strengths and limitations of study)

- > 6) In calculating incident multimorbidity it wasn't clear how death or loss
- > to follow-up during the interval was treated. It seems from the change in
- > sample size from baseline to TP2 that there was fairly significant attrition.
- > Would it be possible to utilize the yearly hospital data using survival
- > analysis and include the competing risk of death?
- >

Our study examines the subsequent risk of hospitalisation within a community population broadly comparable to the UK population. We use number of hospital admissions and length of stay over a fixed period which differs from survival analysis which uses age at the earliest admission and cannot account for multiple events over a long period. Record linkage to national database was performed annual from 1999 to 2019 and since participant migration is rare there is almost no loss to follow-up (methods, paragraph 1). The large reduction in numbers of those attended a baseline and those attending TP2 examinations is mainly due to participants choosing not to attend the later examination.

- > 7) The rationale for the 5 models should be described in the methods section
- > as well as the assessment of model fit.

The rationale for the model covariates used in table 3 and 4 is briefly explain in the results but are examined in more detail in our previous papers which are cited.

- > 8) In the discussion the authors suggest that the CCI is the most widely used
- > index of multimorbidity and can be considered a de facto standard. Perhaps
- > this should be qualified by the authors as their definition of multimorbidity
- > is not the standard definition (based on the number of chronic conditions)
- > used by most researchers

The word de facto has been replace by "common" and the reasons for choosing the CCI are now described more clearly in the discussion.

- > 9) The last paragraph of the discussion prior to the conclusion was very
- > helpful to provide context for examining the two periods. It may be helpful
- > to move some of this motivation to the introduction.

We agree that the introduction was not sufficiently clear in explaining the motivation for the study and the rationale for the two study periods. The introduction has now been modified to explain this.

VERSION 2 – REVIEW

REVIEWER	Heinz Freisling International Agency for Research on Cancer, France
REVIEW RETURNED	16-Jul-2020

GENERAL COMMENTS	Thank you for the revisions. Please add to the abstract that multivariable logistic regression was used as indicated in your response letter.
---

VERSION 2 – AUTHOR RESPONSE

Reviewer Name: Heinz Freisling

Institution and Country: International Agency for Research on Cancer, France

Thank you for the revisions. Please add to the abstract that multivariable logistic regression was used as indicated in your response letter.

Response: The abstract has been altered and now states that multivariable logistic regression was used. For consistency, the term “multivariable” is now used rather than “multivariate” elsewhere in the manuscript.